



# Diverging future surface mass balance between the Antarctic ice shelves and grounded ice sheet

Christoph Kittel [1], Charles Amory [1,2], Cécile Agosta [3], Nicolas C. Jourdain [3], Stefan Hofer [4], Alison Delhasse [1], Sébastien Doutreloup [1], Pierre-Vincent Huot [5], Charlotte Lang [1], Thiery Fichefet [5], and Xavier Fettweis [1]

[1]Department of Geography, UR SPHERES, University of Liège, Belgium
[2]Laboratoire des Sciences du Climat et de l'Environnement, LSCE-IPSL, CEA-CNRS-UVSQ, Université Paris-Saclay, Gif-sur-Yvette, France
[3]Univ. Grenoble Alpes/CNRS/IRD/G-INP, IGE, Grenoble, France
[4]Department of Geosciences, University of Oslo, Oslo, Norway
[5]Earth & Climate, Earth and Life Institute, Catholic University of Louvain, Louvain-la-Neuve, Belgium

**Correspondence:** Christoph Kittel (ckittel@uliege.be)

**Abstract.** The future surface mass balance (SMB) will influence the ice dynamics and the contribution of the Antarctic ice sheet (AIS) to the sea-level rise. Most of recent Antarctic SMB projections were based on the 5th phase of the Coupled Model Intercomparison Project (CMIP5). However, new CMIP6 results have revealed a +1.3°C higher mean Antarctic near-surface temperature than in CMIP5 at the end of the 21st century enabling estimations of future SMB in warmer climates.

Here, we investigate the AIS sensitivity to different warmings with an ensemble of four simulations performed with the polar regional climate model MAR forced by two CMIP5 and two CMIP6 models over 1981–2100. Statistical extrapolation allows us to expand our results to the whole CMIP5 and CMIP6 ensembles. Our results highlight a contrasting effect on the future grounded ice sheet and the ice shelves. The SMB over grounded ice is projected to increase as a response to stronger snowfall, only partly offset by enhanced meltwater runoff. This leads to a cumulated sea-level rise mitigation (i.e. an increase in surface

mass) of the grounded Antarctic surface by 5.1 ± 1.9 cm sea-level equivalent (SLE) in CMIP5-RCP8.5 and 6.3 ± 2.0 cm SLE in CMIP6-ssp585. Additionally, the CMIP6 low-emission ssp126 and intermediate-emission ssp245 scenarios project a stabilised surface mass gain resulting in a lower mitigation to sea-level rise than in ssp585. Over the ice shelves, the strong runoff increase associated with higher temperature is projected to lower the SMB with a stronger decrease in CMIP6-ssp585 compared to CMIP5-RCP8.5. Ice shelves are however predict to have a close-to-present-equilibrium stable SMB under CMIP6

ssp126 and ssp245 scenarios. Future uncertainties are mainly due to the sensitivity to anthropogenic forcing and the timing of the projected warming. While ice shelves should remain at a close-to-equilibrium stable SMB under the Paris Agreements, MAR projects strong SMB decrease for an Antarctic near-surface warming above +2.5°C limiting the warming range before potential irreversible damages on the ice-shelves. Finally, our results reveal the existence of a potential threshold (+7.5°C) that leads to a lower grounded SMB increase. This however has to be confirmed in following studies using more extreme or longer

future scenarios.



## 1 Introduction

The surface mass balance (SMB) of the Antarctic ice sheet (AIS) is the resultant of accumulation through snowfall and ablation through surface erosion, sublimation and runoff. Positive (negative) SMB values reflect a mass gain (loss) at the surface of the ice sheet. The AIS currently loses mass mainly by ice discharge and basal melting. The difference between SMB and ice

discharge determines the sea-level rise contribution of the AIS. Due to the large amount of grounded ice, the AIS is the largest potential contributor among the cryosphere (58 m sea-level equivalent (SLE), Fretwell et al., 2013; Morlighem et al., 2020). Although not directly contributing to sea-level variations, relatively-flat and large ice shelves, i.e. the floating extensions of the ice sheet, nevertheless influence the ice dynamics by retaining the ice over the grounded continent that flows under the force of gravity toward the ocean. This buttressing effect first limits glacier-flow acceleration and then control ice discharge (e.g.,

Rignot et al., 2004; Dupont and Alley, 2005; Gudmundsson, 2013; Fürst et al., 2016).

Since the 2000s, the Antarctic ice sheet has been losing mass at an accelerating rate mainly due to an increased ice discharge in the West AIS (Shepherd et al., 2018), itself caused by the acceleration of outlet glaciers in response to basal (ocean) melt thinning the ice shelves and reducing their buttressing effect (Paolo et al., 2015; Gardner et al., 2018; Rignot et al., 2019). Despite stable surface melt rates since 1979 (Kuipers Munneke et al., 2012), atmospheric conditions through intense melt

events can lead to meltwater ponding at the surface of ice shelves, increasing their potential for hydrofracturing (Scambos et al., 2000; van den Broeke, 2005). The resulting ice-shelf collapses over the Antarctic Peninsula then caused enhanced ice discharge (Scambos et al., 2004, 2014), highlighting the important role of atmosphere-surface interactions in the AIS stability, likely to become even more important in the context of global warming.

With increasing temperatures, more surface mass gain is expected over the AIS as a result of an increase in precipitation

(Palerme et al., 2017; Gorte et al., 2019). Frieler et al. (2015) suggested an increase in accumulation linked to air temperature of $\sim 6\,\%\,°C^{-1}$ that is confirmed by SMB reconstructions from ice cores over the 20th century (Medley et al., 2018; Medley and Thomas, 2019), but not retrieved in recent (too short) SMB reconstructions (Van Wessem et al., 2018; Agosta et al., 2019; Mottram et al., 2020) due to the internal climate variability determining precipitation pattern (Previdi and Polvani, 2016). For moderate warming, increase in snowfall is likely to outpace increased losses through ablation and especially runoff making

the Antarctic SMB the only future mitigating contributor to sea-level rise (Krinner et al., 2007; Agosta et al., 2013; Ligtenberg et al., 2013; Lenaerts et al., 2016; Garbe et al., 2020). Melt increase under the high emissions pathway by 2100 is however projected to be large enough to enhance ice-shelf collapses (Trusel et al., 2015; Donat-Magnin et al., 2020). The future of ice shelves experiencing more snowfall that can enable the snowpack to absorb more liquid water, is still uncertain even if the firn air content should decrease (Ligtenberg et al., 2014; Donat-Magnin et al., 2020), suggesting an increased risk of hydrofracturing

and collapse (Kuipers Munneke et al., 2014).

The most recent projections of the Antarctic SMB are based on global climate models of the 5th phase of the Coupled Model Intercomparison Project (CMIP5) (Taylor et al., 2012), whereas new climate projections are now available through CMIP6 (O'Neill et al., 2016). Under the highest emission scenario, projections for the AIS annual mean near-surface temperature in 2100 are +1.3°C higher in CMIP6 models than in CMIP5 models (Fig. 1). However, using these climate models outputs



directly to study the evolution of the SMB often involves several compromises: (i) their resolution remains too coarse to correctly represent the steep margins of the ice sheet or the peripheral ice shelves (Seroussi et al., 2020) and (ii) they do not account properly for important physical processes of polar regions, in particular those related to the stable boundary layer and snow metamorphism, snowmelt, albedo feedbacks, and refreezing in the snowpack (Lenaerts et al., 2016; Favier et al., 2017). This partly explains why the SMB derived from ESMs has often been roughly approximated as precipitation minus evaporation

even for projections (e.g., Palerme et al., 2017; Favier et al., 2017; Gorte et al., 2019; Seroussi et al., 2020) or included a runoff computed from non-polar-oriented models (Golledge et al., 2015; Nowicki et al., 2020; Garbe et al., 2020), although a few exceptions exist (e.g., Lenaerts et al., 2016; Sellar et al., 2019).

   Dynamical downscaling of ESMs (hereafter designating both global climate models and new-generation Earth System Models without any consideration of the model sophistication to represent the carbon cycle or cloud-aerosol interactions) with

polar-oriented regional climate models (RCMs) offers an alternative to address not only the issue of coarse spatial resolution, but also more importantly to more robustly evaluate changes in mass and energy fluxes at the ice-sheet surface (e.g., Fyke et al., 2018; Lenaerts et al., 2019; Fettweis et al., 2020). This is why we propose here to use the polar-oriented RCM MAR, widely used over the AIS (e.g., Kittel et al., 2018; Agosta et al., 2019; Wille et al., 2019) to downscale an ensemble of 4 different ESMs from the CMIP5 and CMIP6 exercises, selected to cover a wide range of near-surface warming (+3.2°C to +8.5°C

over the Antarctic ice sheet during the 21st century and then statistically extrapolate our results to the full CMIP5 and CMIP6 ensembles. This study therefore aims to (1) quantify the surface response of the AIS to warmer climates, and more specifically the different responses of the grounded ice and ice shelves using both new scenarios and an adapted representation of polar processes, (2) discuss the evolution of individual SMB components including future runoff ablation that can significantly compensate for mass gained through snowfall, (3) assess the future contribution (and related uncertainties) of the grounded

Antarctic SMB to sea-level rise and the future state of the peripheral ice shelves using all the CMIP5 and new CMIP6 models with different emissions scenarios by extrapolating RCM-derived SMB projections.

## 2   Methods

### 2.1   The regional atmospheric model MAR

MAR is a polar-oriented regional climate model abundantly used to study the Antarctic (e.g., Amory et al., 2015; Kittel et al.,

2018; Agosta et al., 2019) and Greenland (e.g., Fettweis et al., 2017; Hofer et al., 2017; Delhasse et al., 2019) ice sheet climates. MAR is a hydrostatic model relying on the primitive equations described in Gallée and Schayes (1994). The model includes a cloud microphysics module solving conservation equations for five water species: snow particles, cloud ice crystals, rain drops, cloud droplets and specific humidity (Gallée, 1995). Airborne particles can be advected vertically from one atmospheric layer to another and notably contribute through sublimation to the heat and moisture budget of the atmosphere (Agosta et al.,

2019). The radiative transfer scheme is adapted from ECMWF ERA-40 reanalysis (Morcrette, 2002). The transfer of mass and energy between the surface and the atmosphere is simulated in the 1-D surface scheme SISVAT (Soil Ice Snow Vegetation Atmosphere Transfer; De Ridder and Gallée, 1998) module, which consists of soil and vegetation (De Ridder and Schayes,



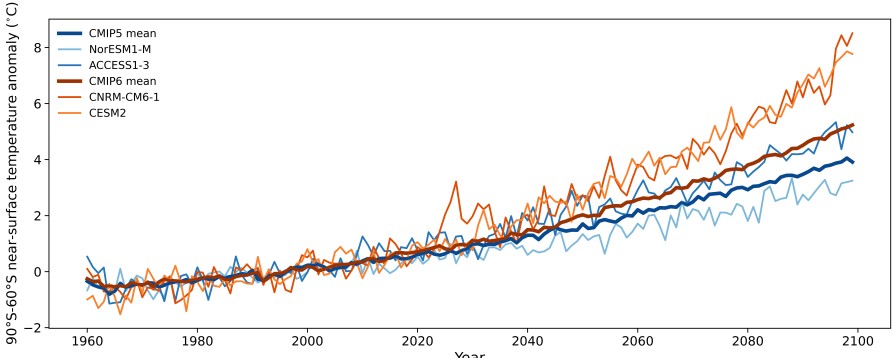

**Figure 1.** Time series of the 90°S–60°S annual near-surface temperature anomaly (°C) between 1960 and 2100 compared to the present reference period (1981–2010) from both the extreme high-emission scenarios RCP8.5 and ssp585. The thick blue and red lines represents the mean annual warming from 28 CMIP5 and 34 CMIP6 ESMs. Thinner orange and blue lines are for ESMs selected as boundary conditions for our regional climate model MAR: CNRM-CM6-1 and CESM2 (CMIP6, ssp585) and NorESM1-M and ACCESS1-3 (CMIP5, RCP85).

1997), snow (Gallée and Duynkerke, 1997; Gallée et al., 2001) and ice (Lefebre et al., 2003) sub-modules. The latter two are originally based on the snow model CROCUS (Brun et al., 1992). The dynamical snow and ice components represent

snow properties and metamorphism across 30 snow/firn/ice layers resolving the 20 first meters of snow/ice. SISVAT solves the surface energy budget using excess in energy to melt the snow. Each snow/firn layer has a maximum water retention of 5%, while the remaining liquid water - coming from rainfall or surface meltwater - can freely percolate downward as long as the underlying snow density does not reach a close-off density of $830 \ \mathrm{kg \ m^{-3}}$. Remaining liquid water beyond the snowpack saturation is converted into surface runoff, meaning that in the absence of water-routing hydrologic scheme, all surface water

that could potentially form melt ponds is considered as runoff, i.e., is lost by the ice sheet. Snow/ice surface albedo varies as a function of the optical properties of snow, the presence of bare ice or liquid water, the snow depth over ice and clouds (Tedesco et al., 2016) with a maximum value of 0.94 for fresh snow and a minimum value of 0.55 for bare ice over Antarctica.

   In this study, we used the latest MAR version (3.11), hereafter called MAR. The latest updates in MAR improve the cloud lifetime, the model stability, its computational efficiency and reduce the dependency to the model time-step. Several other

improvements have also been made in MAR and relative to the previous model versions used over Antarctica (Agosta et al., 2019) and are detailed below:

a. Inclusion of rock outcrops in the ice-sheet mask enabling potential feedbacks between low-albedo exposed rocks (0.17 in MAR) and enhanced snow melting (e.g., Kingslake et al., 2017) around pixels partially composed of rocks. This also resulted in the ice mask being enlarged at the margins while reducing the ice sheet area. SISVAT computes the different

exchanges for the rock surface separately from the snow/firn/ice covered part and then weight-aggregates them according to the proportion of each sub-grid cell.





b. Addition of a vertical atmospheric level (from 23 to 24) for a CPU-allocation reason (better parallelisation along the vertical axis). Note that MAR near-surface results are not sensitive to the additions of more atmospheric levels (Amory et al., 2020(in preperation)).

c. Modification of the fresh falling snow density, now computed as a function of the $10\,\mathrm{m}$ wind speed $\mathrm{ws}_{10}$ ($\mathrm{m\,s^{-1}}$) only:

$$\rho_{\mathrm{s}} = 200 + 32\,\mathrm{ws}_{10}, \tag{1}$$

with minimum and maximum values fixed to 300 and $400\,\mathrm{kg\,m^{-3}}$ in accordance with observations (Agosta et al., 2015, Table S2) and the new developments into the drifting-snow scheme (Amory et al., 2020(in preperation)).

The Antarctic topography, and ice/rock fraction are computed from the 1 km resolution digital elevation model Bedmap2

(Fretwell et al., 2013). The ice mask is fixed and cannot evolve, meaning that changes in ice extent following for instance an ice-shelf collapse are not represented. The same is true for surface elevation that is assumed to remain constant in the absence of ice dynamics and evolving topography. Therefore, feedbacks between the ice sheet geometry and the atmosphere are not taken into account in our simulations. Finally, as the drifting-snow scheme (Amory et al., 2020(in preperation)) was still under development when we performed our simulations, it was not activated in this study.

**2.1.1 Selection of ESMs**

The selection of ESMs that were dynamically downscaled by MAR was based on their ability to 1) represent the current climate (air temperature and humidity, sea surface conditions, and large-scale circulation) around the AIS and 2) diversify the projected changes during the 21st century. These criteria ensure on one hand, that the ESM biases will not have a prejudicial effect on the projections since the present state determines future biases (Agosta et al., 2015; Krinner and Flanner, 2018) and on the

other hand that we assess the AIS response to a wide range of projected temperature increases for a better quantification of the future uncertainties. We therefore selected ESMs by comparing them to the ECMWF reanalysis ERA5 (Hersbach et al., 2020) over the recent "historical" period (1980–2004) following the method defined in Agosta et al. (2015) and Barthel et al. (2020) for CMIP5, extended here to CMIP6 and applied only to the Antarctic atmosphere.

Large-scale forcing models were chosen among the CMIP5 and CMIP6 ESMs with 6-hourly outputs available needed

by MAR. CMIP6 models rely on an improved and more sophisticated representation of the global climate system than CMIP5. They incorporate better coupling between the different components of the Earth system, improved present- and better-constrained future concentrations scenarios of long-lived greenhouse gases and aerosols (Eyring et al., 2016; O'Neill et al., 2016). Additionally, most CMIP6 ESMs are also run on a higher spatial resolution. First analyses of the CMIP6 results revealed higher equilibrium climate sensitivity in this new-generation models (Mauritsen et al., 2019; Voldoire et al., 2019; Zelinka et al.,

2020; Meehl et al., 2020; Wyser et al., 2020), suggesting warmer future climates, while based on similar future scenarios in terms of global radiative forcing. However, this higher climate sensitivity is potentially not supported by paleo-climate records (Zhu et al., 2020). We therefore also included models from the CMIP5 dataset, some of which show a good agreement with reanalyses over the current Antarctic climate (Agosta et al., 2015; Palerme et al., 2017). Finally, we only chose the scenarios of



large greenhouse gas emissions from CMIP5 (RCP8.5) and its updated version in CMIP6 (ssp585) in order to obtain stronger
warming signals. These two scenarios have an equivalent global radiative forcing of $+8.5\,\mathrm{W\,m^{-2}}$ by 2100, but differ in how
the anthropogenic forcing is split between individual drivers of global warming (O'Neill et al., 2016).

We selected two models from the CMIP5 ensemble, ACCESS1.3 and NorESM-M, and two from CMIP6, CNRM-CM6-1
and CESM2. The Antarctic (90°S–60°S) near-surface warming they produce for RCP8.5 (CMIP5) and ssp585 (CMIP6) is
shown in Fig. 1. ACCESS1.3 (Bi et al., 2013; Dix et al., 2013) is the model that best represents the present Antarctic climate
compared to ERA-Interim (Agosta et al., 2015), and is also among the best models when compared to ERA5 (Agosta et al., in
preparation). This ESM has a near-surface Antarctic warming close to the CMIP6 multi-model mean (+5°C, ). NorESM1-M
(Bentsen et al., 2013; Iversen et al., 2013) projects a weaker Antarctic atmospheric warming (+3.2°C, Fig. 1) but a stronger
ocean warming (Barthel et al., 2020). CNRM-CM6-1 (Voldoire et al., 2019) correctly represents the present Antarctic climate
and was among the first models available in the CMIP6 data base. This model also enables to assess the AIS response to an
extreme Antarctic warming (+8.5°C) since it is the warmest model over the AIS among the CMIP5 and CMIP6 databases at
the end on the 21st century. CESM2 (Danabasoglu et al., 2020) has a lower score than half of the CMIP5 and CMIP6 models
compared to ERA5 (Agosta et al., in preparation). Despite its modest ranking, it was chosen due to its relatively detailed
representation of polar oriented processes, early availability, and the frequent use of this model and its earlier version to study
the AIS (e.g., Lenaerts et al., 2016; Fyke et al., 2017; Medley et al., 2018; Nowicki et al., 2020). Its projected warming (+7.7°C)
is close to the mean warming projected by CNRM-CM6-1. From this perspective, selecting both CESM2 and CNRM-CM6-1
does not maximise the warming range covered and prevents our selected ESM ensemble to be representative of the mean
CMIP5 and CMIP6 warming. Yet, it enables us to assess the AIS response (and uncertaities) related to the strong warming that
is only projected by a few ESMs.

### 2.1.2 Experiments

MAR is forced by 6-hourly large-scale forcing fields at its atmospheric lateral boundaries (pressure, wind, specific humidity,
and temperature), at its sea surface (sea ice concentration and sea surface temperature), as well as at the top of the troposphere
(wind and temperature). We forced MAR with the selected ESMs over 1976–2100 (Sect. 2.1.1), and the first five years (1975–
1980) were discarded as spinup. The simulations are called MAR(ACCESS1.3), MAR(CESM2), MAR(CNRM-CM6-1), and
MAR(NorESM1-M) hereafter. We used the same intermediate spatial resolution (35 km) as in (Agosta et al., 2019) and
(Mottram et al., 2020) as a computation time compromise to run the model with multiple forcings over the 20th and the 21st
centuries. In order to assess the quality of the downscaling over the present climate, we also forced MARv3.11 by the ERA5
reanalysis (MAR(ERA5) hereafter). This comparison between MAR forced by the different ESMs, as well as an evaluation of
MAR(ERA5), is available as supplementary material. It shows that MAR(ERA5) performs similarly as MARv3.10 forced by
ERA-Interim, which was among the best simulations in terms of both present Antarctic near-surface climate and SMB in the
recent evaluation conducted by Mottram et al. (2020). We refer to Suplement S1 for more details about the comparison and
evaluation of MARv3.11 in terms of near-surface climate, melt and SMB.





In this study, we have chosen to define the reference period of the present climate as 1981–2010. This 30-year reference period coincides with the availability of reanalyses and is a compromise between the end of the historical scenarios, which last until 2004 for CMIP5 and 2014 for CMIP6. Furthermore, Mottram et al. (2020) showed that this period is characterised by a
relatively stable SMB over Antarctica.

## 3 Results

Our ESMs-based experiments closely reproduce the SMB and near-surface climate of MAR(ERA5) over the historical period (Supplement S2). The anomalies of the annual mean SMB modelled by MAR forced by each ESM compared to MAR(ERA5) are lower than the interannual variability (i.e, one standard deviation) of the SMB simulated by MAR(ERA5) over the historical
period, suggesting that the biases are not significant. Overall, MAR(ACCESS1.3) has the best representation of the Antarctic SMB over the current climate (mean bias: -3 Gt yr$^{-1}$, spatial rmse: 59 kg m$^{-2}$ yr$^{-1}$), while MAR(CESM2) is the least accurate (mean bias: -25 Gt yr$^{-1}$, spatial rmse: 90 kg m$^{-2}$ yr$^{-1}$). We refer to Supplement Sect. S2 to more details about the evaluation of our experiments. The results of our experiments over the current climate are consistent with the ranking of the ESMs given by Agosta et al. (2015), Barthel et al. (2020), and Agosta et al. (in preparation). This highlights the importance of selecting
ESMs that correctly represent the historical climate around Antarctica as they strongly controls present biases independently of the capacity of the RCM to improve ESMs results.

Our projections of the Antarctic SMB show a trend towards surface mass gains by the end of the 21st century (Fig. S6). MAR simulations forced by the high-emission scenarios ssp585 and RCP8.5 suggest a generally higher Antarctic SMB (including ice shelves) during 2071–2100 than for 1981–2010, with positive anomalies between +257 Gt yr$^{-1}$ for MAR(CNRM-CM6-1)
and +505 Gt yr$^{-1}$ for MAR(CESM2). The projections reveal a spread of 248 Gt yr$^{-1}$, i.e., almost a factor two between the lowest and the highest increase in SMB. Such a high amplitude highlights the importance of using multiple models for a better assessment of the uncertainties when discussing the future state of the Antarctic SMB throughout the 21st century.

### 3.1 Regional changes

Using Antarctic-integrated values however hides two distinct signals. The diverging trajectories of SMB over grounded versus
floating ice (Fig. 2) suggest contrasted processes at play. In the rest of this manuscript, we will therefore discuss separately the ice shelves and the grounded ice sheet. This distinction is also justified by the direct equivalent between grounded-ice mass change and mean sea-level variations, whereas ice shelves do not directly contribute to sea-level variations even if their surface processes (such as hydrofracturing) are of crucial importance for the ice-sheet dynamics and therefore the Antarctic mass balance evolution. The locations mentioned hereafter are illustrated in Fig. S7.

### 3.1.1 Grounded ice sheet

The grounded Antarctic SMB is projected to increase by +347 Gt yr$^{-1}$ (MAR(NorESM1-M)) to +745 Gt yr$^{-1}$ (MAR(CESM2)) from 1981–2010 to 2071–2100 (Table 1). Our simulations suggest large (up to more than twice the present - natural - inter-





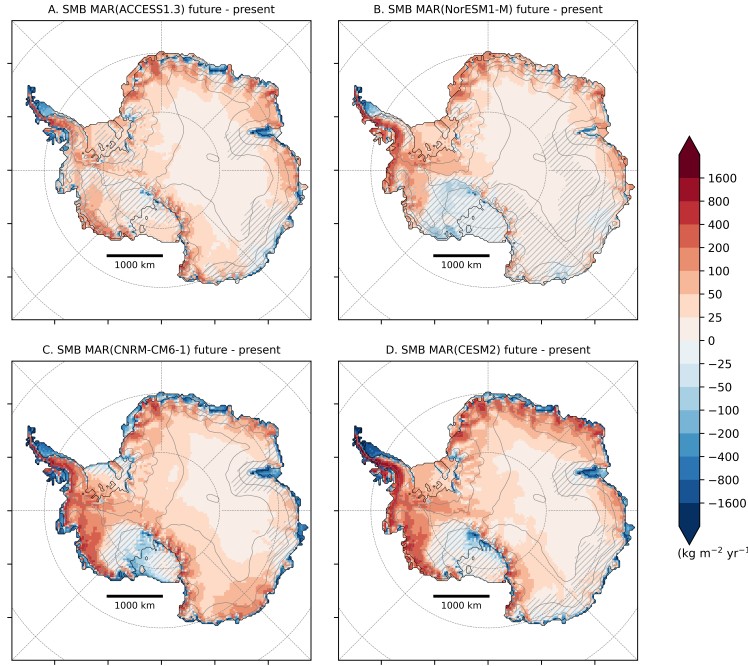

**Figure 2.** SMB changes $(\mathrm{kg\,m^{-2}\,yr^{-1}})$ between 2071–2100 and 1981–2010 as modelled by MAR forced by ACCESS1-3 (A), NorESM1-M (B), CNRM-CM6-1 (C), and CESM2 (D). Locations where future changes are smaller than the (natural) interannual variability over the present climate (interannual standard deviation) are hatched.

annual variability) positive SMB anomalies in West Antarctica (Marie Byrd and Ellsworth Land) and over the mountainous regions of the Antarctic Peninsula (Fig. 2). The situation in East Antarctica is more contrasted. The increase is significant (i.e,

larger than the interannual variability over 1981–2010) in Queen Mary Land and high-elevation plateaus, while George V Land, Adelie Land and Wilkes Land are projected to have a weak increase in SMB for all the simulations, except MAR(CNRM-CM6-1) which suggests a strong increase there.

From 2015 onwards, the grounded SMB increases in all our MAR simulations (Fig. 4A). Large differences between projections appear around 2040–2050 when MAR(CESM2) and MAR(CNRM-CM6-1) suggest the strongest increase after 2050

and 2065 respectively while MAR(NorESM1-M) and MAR(ACCESS1.3) show a substantial increase at the end of the 21st century. Finally, only MAR(CNRM-CM6-1) suggests a SMB decrease beyond 2095.

The grounded SMB trend is mainly dominated by an increase in snowfall (Fig. 4B). Increased air moisture content associated to higher air temperatures leads to a widespread increase in snowfall over the AIS, explaining most of the positive SMB anomalies. This increase is stronger where air masses saturate as they adiabatically cool when rising with the topography (Agosta

et al., 2013; Ligtenberg et al., 2013). Figure 3 shows that the largest increase occurs in West Antarctica, where the accumulation by snowfall already is the highest over the present climate. Although more snowfall can be expected over most of the AIS in a warmer climate (Palerme et al., 2017), some parts of the Antarctic grounded ice sheet show negative anomalies. This decrease

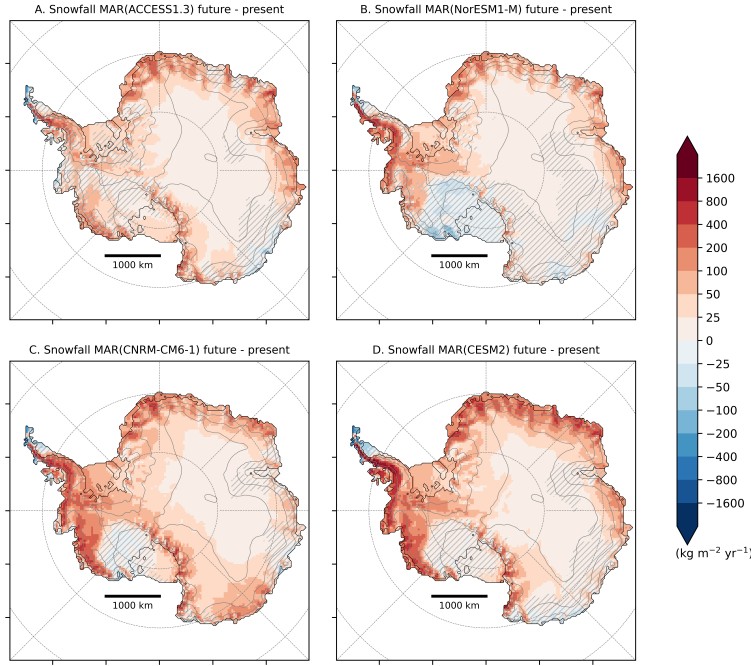

**Figure 3.** Snowfall changes $(\mathrm{kg\,m^{-2}\,yr^{-1}})$ between 2071–2100 and 1981–2010 as modelled by MAR forced by ACCESS1-3 (A), NorESM1-M (B), CNRM-CM6-1 (C), and CESM2 (D), using ssp585 and RCP8.5. Locations where changes are smaller than the (natural) interannual variability of the present climate (interannual standard deviation) are hatched.

in snowfall affects areas such as inland of Marie Byrd where the SMB consequently decreases. This strong snowfall increase over the peripheral slopes associated afterwards with an inland reduction could result from enhanced condensation over the

marginal slopes reducing moisture intrusion and snowfall formation inland (Kittel et al., 2018). Although this effect may be present in our projections, Figure S8(b) also reveals a deepening of the Amundsen Sea Low enhancing moisture advection towards the Antarctic peninsula in MAR(NorESM1-M). This deepening projected by NorESM-M especially occurs in winter (Raphael et al., 2016) and result from rising greenhouse-gas emissions (Hosking et al., 2016; Raphael et al., 2016).

Snowfall increase in response to higher air temperatures also competes with a subsequent increase in runoff over the

grounded ice margins (Fig. 4E). Although runoff amounts are negligible in the present climate and the increase in runoff is lower than the increase in snowfall, the future runoff contribution could compensate up to 34% of the snowfall increase in MAR(CNRM-CM6-1) over 2071–2100, questionning the use of Precipitation-Evaporation in pace of SMB used in earlier studies (e.g., Palerme et al., 2017; Favier et al., 2017; Gorte et al., 2019). Other surface mass flux components such as rainfall (Fig. 4G), deposition and sublimation are not projected to contribute significantly to SMB changes.

From 1981 to 2100, our results suggest a grounded cumulative contribution of -3.7 cm, -5.8 cm, -8.1 cm and -10.6 cm SLE for MAR(NorESM1-M), MAR(ACCESS1.3), MAR(CNRM-CM6-1) and MAR(CESM2) respectively. Given that all these



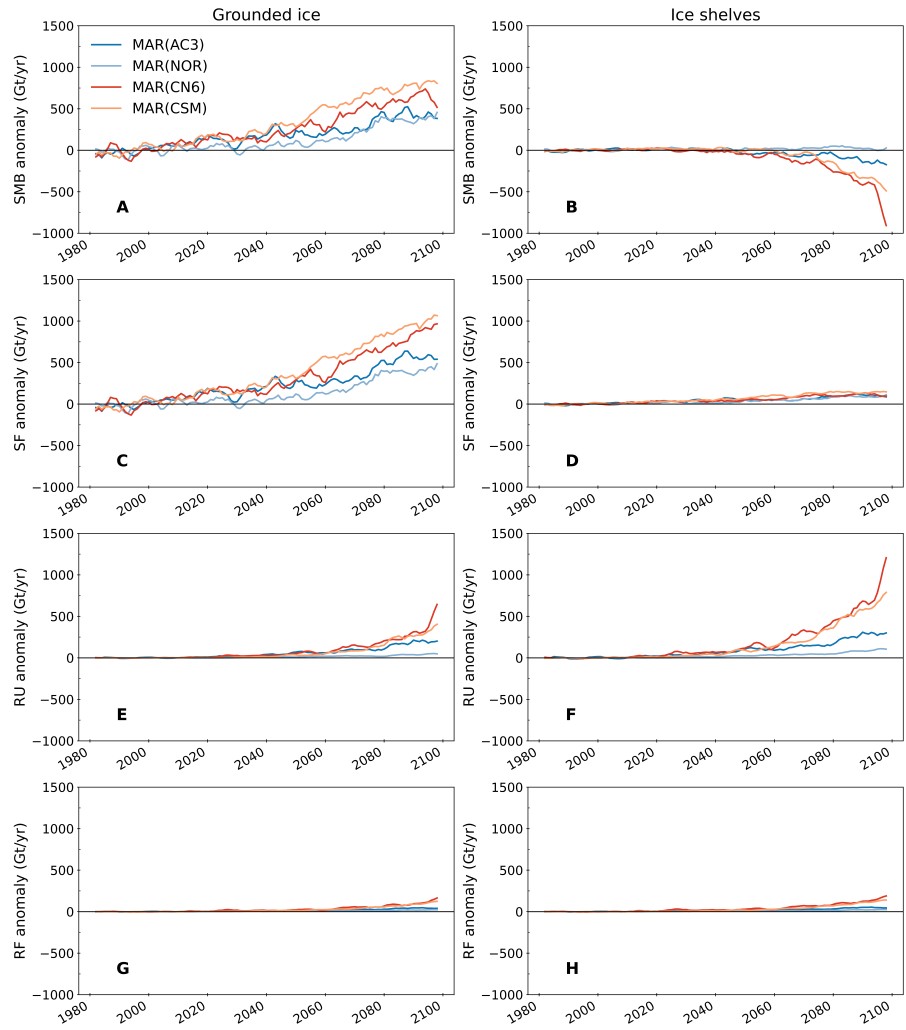

**Figure 4.** Time series of the integrated annual SMB (A, B), snowfall (C, D), runoff (E, F) and rainfall (G, H) anomalies (Gt yr$^{-1}$) over the Antarctic grounded ice (A, C, E, G) and the Antarctic ice shelves (B, D, F, H) from 1980 to 2100 simulated by MAR forced by RCP8.5 or ssp585 scenarios from ACCESS1-3 (blue), NorESM1-M (light blue), CNRM-CM6-1 (red), and CESM2 (orange) compared to the 1981–2010 reference period. A running average of 5 years was applied to the original time series for better readability. Sublimation and surface melt changes are shown in Fig.S9

projections are obtained from similar anthropogenic forcing, this demonstrates the necessity of using several ESMs to evaluate the Antarctic contribution to the sea-level rise in high-emission scenarios at the end of the 21st century.





### 3.1.2 Ice shelves

The SMB evolution over the ice shelves shows more uncertainties depending on the forcing ESM. It remains close to the present-day values in MAR(NorESM1-M), while it strongly decreases after 2075 in the other simulations (Fig. 4B). All the MAR simulations agree on a significant SMB decrease over the ice shelves on the lee side (eastern) of the Northern Antarctic Peninsula and near Amery's grounding line (Fig. 2). With the exception of MAR(NorESM1-M), our projections also suggest a strong SMB decrease over the ice shelves on the windward side of the Northern Peninsula and over a majority of the ice

shelves in Wilkes Land and in Queen Maud Land. Only MAR(CNRM-CM6-1) reveals widespread negative SMB anomalies over all the small Antarctic peripheral ice shelves. The Ronne-Filchner Ice Shelf is expected to have an increase in SMB, even in MAR(CNRM-CM6-1) except in the vicinity of the ocean. Our simulations suggest diverging responses over the Ross ice shelf, positive in MAR(ACCESS1.3) and MAR(CESM2), negative in MAR(CNRM-CM6-1) and MAR(NorESM1-M). The Ross ice shelf illustrates the large uncertainties related to the different model forcings on the future SMB over the Antarctic ice

shelves until 2100.

   MAR suggests an increase in snowfall over ice shelves (between +83 Gt yr$^{-1}$ and +139 Gt yr$^{-1}$) regardless of the forcing ESM, but also a significant increase in rainfall (+18 Gt yr$^{-1}$ to 108 Gt yr$^{-1}$) (Table 1). The increase in snowfall over the ice shelves is however weaker than the increase over the grounded margins, suggesting a stronger saturation of air masses when lifted over the ice-sheet slope (Fig. 3). Over the period 2071–2100, rainfall anomalies can be as large as snowfall anomalies

on the ice shelves, or even outpace the increase in snowfall in MAR(CNRM-CM6-1), where snowfall is projected to decrease at the very end of the century (Fig. 4H). The warmer air also induces a conversion of snowfall into rainfall over the Antarctic Peninsula, where the total precipitation is projected to increase despite an increasing fraction falling as rain. Snowfall also decreases over the Ross Ice Shelf in MAR(NorESM1-M) due to a pronounced intensification of the Amundsen Sea Low system bringing more moisture towards the Peninsula and less over the Ross Ice Shelf (Fig. S8b), which reduces SMB over

this area.

   Higher air temperature also causes a significant increase in surface melt. Repeated years of intense melting, combined with increased rainfall, reduce the firn air content and weaken the snowpack capacity to retain liquid water. This results in large runoff production rates over the ice shelves, except over the Ronne-Filchner due to its more southern position, as displayed in Fig. 5. MAR(NorESM1-M) suggests the lowest increase in runoff (+18 Gt yr$^{-1}$), which is one order of magnitude lower than

for MAR(CNRM-CM6-1) (+558 Gt yr$^{-1}$).

   The amount of runoff projected at the end of the century explains the large changes in SMB over the ice shelves (Fig. 4F). The projected SMB decrease in MAR(CNRM-CM6-1) over the Ross ice shelf results from the larger increase in runoff than in snowfall while the decrease in SMB in the MAR(NorESM1-M) experiment is only attributed to reduced snowfall accumulation. Finally, the sharp runoff increase in MAR(CNRM-CM6-1) starting in 2090 reflects a widespread runoff over nearly all the ice

shelves (Fig. 5).



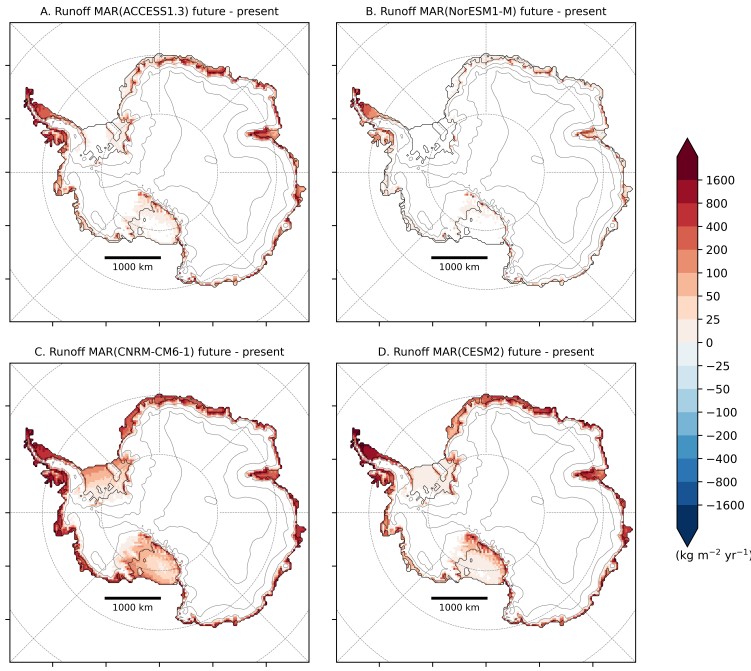

**Figure 5.** Changes in ruonff production ($\mathrm{kg\,m^{-2}\,yr^{-1}}$) between 2071–2100 and 1981–2010 as modelled by MAR forced by ACCESS1-3 (A), NorESM1-M (B), CNRM-CM6-1 (C), and CESM2 (D). Locations where changes are smaller than the (natural) interannual variability of the present climate (interannual standard deviation) are hatched.

**Table 1.** Integrated anomalies ($\mathrm{Gt\,yr^{-1}}$) of SMB, snowfall, rainfall, runoff, net sublimation (defined as surface sublimation minus surface deposition), and melt for the grounded ice sheet and the ice shelves over 2071–2100 compared to the present (1981–2010) from RCP8.5 and ssp585 simulations. All the anomalies are larger than the present interannual variability (i.e, standard deviation) of the same simulation and are therefore considered as significant.

| | SMB | Snowfall | Rainfall | Runoff | Net sublimation | Melt |
|---|---|---|---|---|---|---|
| Grounded ice ($11.94\ 10^6\ \mathrm{km^2}$) | | | | | | |
| MAR(ACCESS1.3) | +390 ± 75 | +501 ± 96 | +36 ± 5 | +151 ± 44 | +4 ± 3 | 277 ± 69 |
| MAR(NorESM1-M) | +437 ± 61 | +367 ± 64 | +18 ± 5 | +32 ± 11 | +4 ± 3 | +79 ± 25 |
| MAR(CNRM-CM6-1) | +592 ± 67 | +753 ± 120 | +85 ± 29 | +260 ± 124 | -20 ± 12 | +490 ± 17 |
| MAR(CESM2) | +745 ± 60 | +880 ± 111 | +75 ± 24 | +221 ± 89 | -17 ± 8 | 395 ± 135 |
| Ice shelves ($1.77\ 10^6\ \mathrm{km^2}$) | | | | | | |
| MAR(ACCESS1.3) | -94 ± +44 | +94 ± 17 | +41 ± 9 | +229 ± 62 | +4 ± 1 | +416 ± 93 |
| MAR(NorESM1-M) | +28 ± 14 | +83 ± 14 | 18 ± 15 | +69 ± 23 | +3 ± 1 | +182 ± 51 |
| MAR(CNRM-CM6-1) | -335 ± 190 | +109 ± 12 | +108 ± 34 | +558 ± 227 | -6 ± 4 | 781 ± 220 |
| MAR(CESM2) | -241 ± 127 | +139 ± 8 | +90 ± 28 | +476 ± 162 | -7 ± 3 | +703 ± 179 |



## 3.2 Links with the ESM near-surface temperature

Our projections of the 21st century evolution of the Antarctic SMB yield large spread in SMB for both the Antarctic grounded ice and ice shelves. This spread can mostly be attributed to different warming rates in the forcing ESM, as they show a broad range of warming rates despite a similar radiative forcing due to anthropogenic emissions (Figure 1).

We identify the 30-year periods (different for each ESM) characterised by an Antarctic (90°S–60°S) annual near-surface climate about +2.5°C warmer on average than the climate over the historical period (1981–2010) to compare SMB anomalies resulting from an equivalent warming. This +2.5°C warming corresponds to the strongest 30-year averaged near-surface warming common to all our selected ESMs. The period selected for each ESM is listed in Table S3. Mean SMB anomalies projected by MAR during these periods reveal a very similar spatial pattern between all our experiment. A +2.5°C warming

yields a mostly non-significant increase in SMB over the grounded ice sheet and a weak (negative) change over the surrounding ice shelves (Fig. S10). This comparison at equivalent warming but different 30-year periods shows that the spread in the future SMB is mainly due to the timing and magnitude of the warming projected by the ESMs.

To remove the uncertainty associated to the different warming rates, we associate the future annual anomalies modelled by MAR to annual near-surface temperature anomalies over 90°S–60°S from the forcing ESM. Figure 6 reveals more consistent

projections between all our experiments. Note that associating annual MAR anomalies with ESM temperature anomalies in the free atmosphere (700 or 850 hPa) does not change the comparison (not shown).

Precipitation increases following the Clausius-Clapeyron relation, a weak exponential form that can be approximated as a (nearly) linear relationship for moderate warming over the AIS (Agosta et al., 2013; Frieler et al., 2015; Palerme et al., 2017). The grounded (Fig. S11A) increase is dominated by snowfall anomalies (Fig. 6A) with a weak contribution of rainfall (Fig. 6C).

Over the ice shelves, snowfall is no longer increasing for strong warmings above +7.5°C. A the total increase in precipitation remains also approximately linear (Fig. S11B), an increasing proportion of the potential additional precipitation falls as rain instead of snow for higher temperature over the ice shelves. Under increasing warming, more locations will experience rainfall, melting and runoff. We therefore link rainfall (Fig. 6C,D) and runoff (Fig. 6E,F) anomalies with near-surface temperature anomalies using a quadratic relation reflecting postive feedbacks (Fettweis et al., 2013). Our results suggest that the increase

in rainfall will be stronger than the snowfall increase over the ice shelves for warming above 7.5°C. The integrated increase in runoff is stronger over ice shelves than over the grounded ice, despite lower floating areas. This is mainly explained by the low surface elevation of the ice shelves. Other studies (e.g., Kuipers Munneke et al., 2014; Trusel et al., 2015; Donat-Magnin et al., 2020) also linked an exponential increase in melting with air temperature over the AIS.

Although the dominant signal explaining grounded SMB variations is the snowfall increase, the trend suggest a slowing

or even a lower grounded SMB increase for warmings higher than +7.5°C (Fig. 7). This results from a strong increase in the grounded ice-sheet runoff. However, this grounded-SMB threshold is only supported by MAR(CNRM-CM6-1). Since this warming magnitude is not reached across all our other projections and as CNRM-CM6-1 is the warmest model in the entire CMIP5 and CMIP6 database, it would require longer projections to confirm the confidence of this threshold.





Over the ice shelves, a near-surface temperature increase by more than +2°C results in runoff anomalies larger than pre-
cipitation anomalies, hence leading to negative SMB anomalies (Fig. 7). While ice-shelf collapses could already occur due to
hydrofracturing caused by enhanced surface melt, additional warming beyond this threshold will result in less surface accumu-
lation, or even ice-shelf thinning for the warmings that result in a SMB decrease stronger than 478 $\mathrm{Gt\,yr^{-1}}$ (i.e., the present
SMB simulated by MAR(ERA5)) over the ice shelves). This might induce marine ice-sheet instability and/or enhancing posi-
tive feedbacks between ice dynamics and new damage weakening the ice shelves (Lhermitte et al., 2020).

**4    Discussion**

**4.1    Statistical projections for the CMIP5 and CMIP6 ensemble**

Anomalies in Antarctic SMB and its driving components (precipitation and runoff) are strongly explained by near-surface ESM
temperature anomalies between 90°S–60°S as discussed above (see Sect. 3.2). We therefore propose to reconstruct the SMB
for both the Antarctic grounded ice (Eq. 2) and ice shelves (Eq. 3) using ESM near-surface temperature anomalies:

$$\Delta SMB_{grd} \approx -1.3\,\Delta TAS^2_{90-60S} + 115.4\,\Delta TAS_{90-60S} - 11.1, \tag{2}$$

$$\Delta SMB_{shf} \approx -12.7\,\Delta TAS^2_{90-60S} + 32.1\,\Delta TAS_{90-60S} - 3.1, \tag{3}$$

where $\Delta SMB_{grd}$, $\Delta SMB_{shf}$, and $\Delta TAS_{90-60S}$ represent the SMB anomalies over the grounded ice and ice shelves
(in $\mathrm{Gt\,yr^{-1}}$), and the ESM 90°S–60°S near-surface temperature anomaly (in °C) compared to their respective mean value
over 1981–2010. A more detailed description of the ability of this regression to represent SMB anomalies is presented in
Supplementary Material (Fig S12). Since CNRM-CM6-1 has the strongest Antarctic near-surface warming among all CMIP5
and CMIP6 models, we can use this regression to predict the future SMB in 2100 without any extrapolation outside the warming
range of our projections. However, this implies several hypotheses, such as the absence of strong atmospheric circulation
changes (influencing humidity advection) or a fixed ice surface (topography).

Using Eq. (2) and Eq. (3), we reconstructed the annual Antarctic SMB for all the CMIP5 (RCP8.5) and CMIP6 (ssp126,
ssp245, ssp585) models for which the annual near-surface temperature is available until 2100. The projected SMB anomalies
remain similar until 2040–2050 in all the reconstructions. They then start to diverge and lead to a difference of -1.2cm SLE
(-6.3 ± 2.0 cm SLE in CMIP6-ssp585 vs -5.1 ± 1.9 cm in CMIP5-RCP8.5 cumulated over the period 1981–2100). From the
period 2045–2050, the SMB on the ice shelves starts decreasing in CMIP5-RCP8.5 models, and even more in CMIP6-ssp585
models, with a multi-model-mean difference of 65 $\mathrm{Gt\,yr^{-1}}$ over 2071–2100. A few models nonetheless suggest a steady-state
ice-shelf SMB both in the CMIP5 and CMIP6 ensembles. It should also be noted that the CMIP6-ssp585 spread is much larger
than in CMIP5-RCP8.5, as it ranges from strong negative anomalies (-600 $\mathrm{Gt\,yr^{-1}}$, i.e. lower than present ice-shelf SMB)
to steady state or even slightly positive anomalies on the ice shelves. The CMIP6-ssp585 ensemble-mean value in 2100 is
also nearly outside the spread range of CMIP5-RCP8.5 models highlighting the average stronger SMB decrease in CMIP6-
ssp585. Similarly to what is projected for the Greenland ice sheet (Hofer et al., 2020(in discussion)), the higher equilibrium





**Figure 6.** MAR snowfall (A, B), rainfall (C, D) and runoff (E, H) anomalies ($\mathrm{Gt\,yr^{-1}}$) over the grounded ice (A, C, E) and ice shelves (B, D, H) compared to the annual near-surface temperature anomaly from the forcing ESM between 90°S-60°S (°C). The black regression was computed using all the MAR-ESM anomalies while individual regression are also represented (coloured lines). The regression equation and determination coefficient are mentioned for each scatter plot.





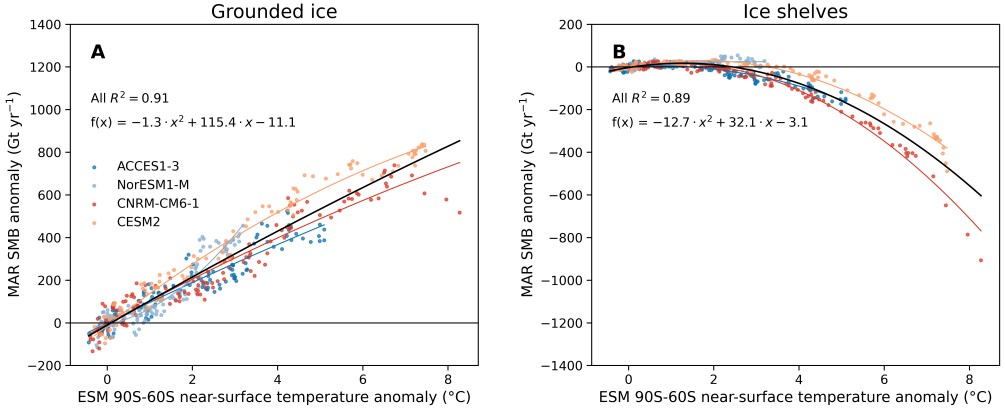

**Figure 7.** MAR SMB anomaly over the grounded ice (A) and ice shelves (B) compared to the annual near-surface temperature anomaly from the forcing ESM between 90°S-60°S (°C). The black regression was computed using all the MAR-ESM anomalies while individual regression are also represented (coloured lines).

climate sensitivity of several CMIP6 models largely explains the differences between the CMIP5 and CMIP6 results. Both the CMIP6-ssp126 and CMIP6-ssp245 scenarios yield a stable SMB (increased over the grounded ice and close to steady state to slightly negative over the ice shelves) after 2050. In cumulative terms, our CMIP6 reconstructions cumulated over the 21st century indicate Antarctic grounded-surface contributions of -3.0 ±1.4 cm SLE for CMIP6-ssp126 and -4.2 ±1.6 cm SLE for CMIP6-ssp245, i.e a lower sea-level rise mitigation than for CMPIP6-ssp585. As described in Sect. 3, a high temperature increase induces higher precipitation rates but also higher runoff over the grounded ice sheet. Figure S13 reveals large spreads in both integrated snowfall and runoff changes. However, as runoff increase partly compensates snowfall increase, the spread in SMB change is strongly reduced compared to the individual components.

## 4.2 Comparison with the ISMIP6-derived SMB

Due to time constraints and computational demands faced by the Ice Sheet Model Intercomparison Project (ISMIP6, Nowicki et al., 2016), future Antarctic projections for forcing ice-sheet models were derived directly from ESMs while, over the Greenland ice sheet MAR was used to downscale ESM projections (Nowicki et al., 2020). However, using ESMs to study the evolution of the SMB often involves several compromises related to their coarse resolution and their low sophistication to represent important physical processes of polar regions. Although RCMs have been believed to add uncertainties in the downscaling product (Nowicki et al., 2016), the significant SMB biases in ESMs over the current climate (e.g., Krinner et al., 2007; Agosta et al., 2015; Lenaerts et al., 2017b; Palerme et al., 2017; Krinner and Flanner, 2018) might be a larger source of uncertainties than the downscaling itself. Therefore, we compare our MAR projections forced by NorESM1-M (RCP8.5), CESM2 and CNRM-CM6-1 (ssp585) to the ISMIP6-derived SMB used to predict the future Antarctic sea-level contribution (Seroussi et al., 2020) by interpolating the 32 km SMB fields built by ISMIP6 on the 35 km MAR grid.





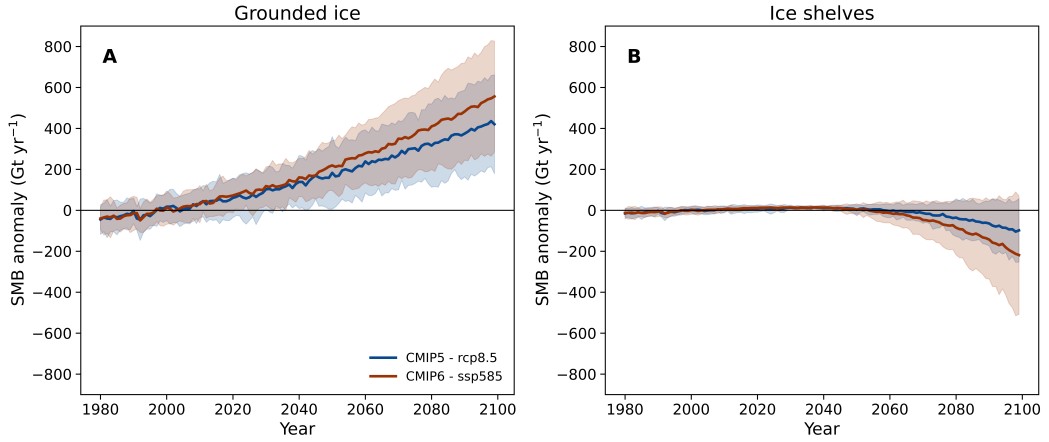

**Figure 8.** Reconstructed SMB anomaly $\mathrm{Gt\,yr^{-1}}$ using CMIP5-RCP8.5 (blue) and CMIP6-ssp585 models (red) over the Antarctic grounded ice (A) and ice shelves (B). Projections are shown using the multi-model mean (solid lines) and the 5 to 95% range, corresponding to $\pm 1.64$ standard deviation, across the distribution of individual models (shading).

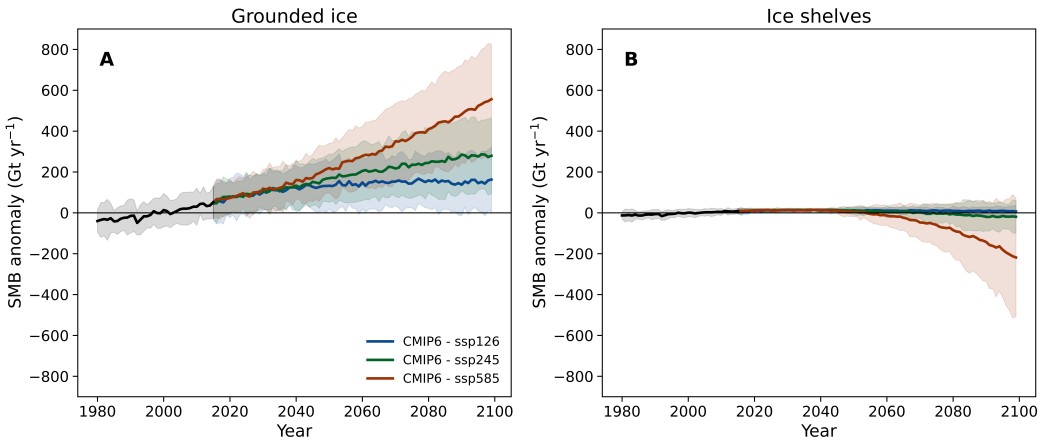

**Figure 9.** Reconstructed SMB anomaly $\mathrm{Gt\,yr^{-1}}$ for the CMIP6 models using the ssp126 (green), ssp245 (green) and ssp585 (red) scenarios over the Antarctic grounded ice (A) and ice shelves (B). Projections are shown using the multi-model mean (solid lines) and the 5 to 95% range, corresponding to $\pm 1.64$ standard deviation, across the distribution of individual models (shading).



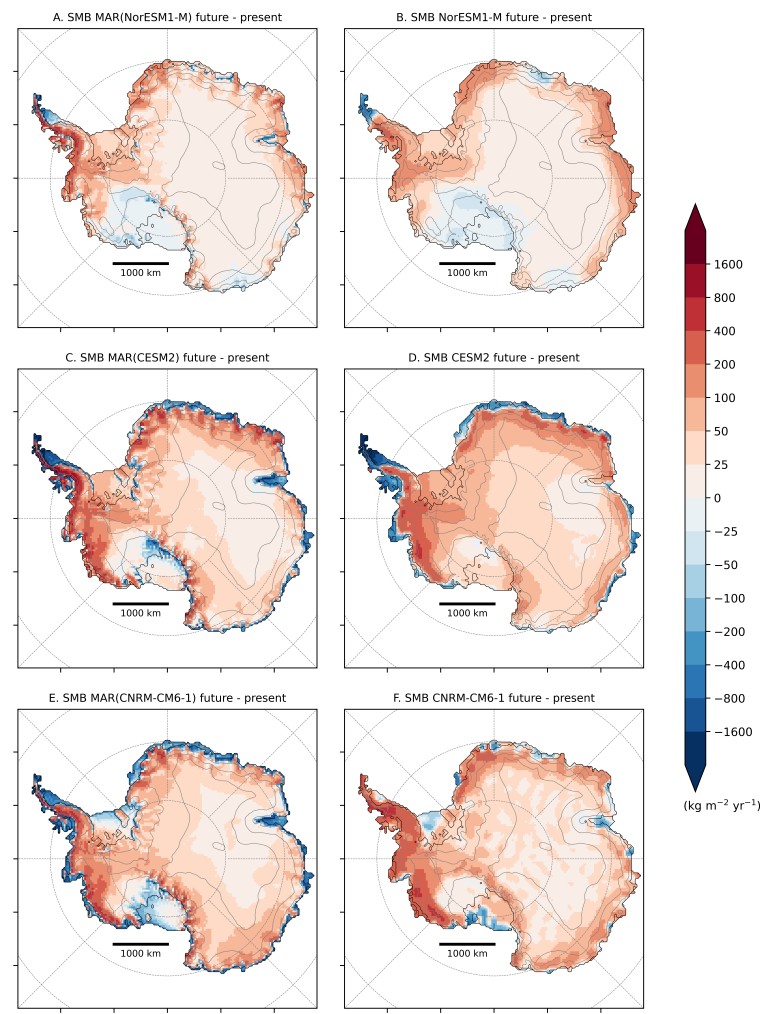

**Figure 10.** Comprison between SMB anomalies between 2081–2100 and 1995–2014 ($\mathrm{kg\,m^{-2}\,yr^{-1}}$) projected by MAR forced by NorESM1-M (A), CESM2(C), CNRM-CM6-1 (E), and the ISMIP6-SMB directly derived from NorESM1-M (B), CESM2 (D) and CNRM-CM6-1 (F).

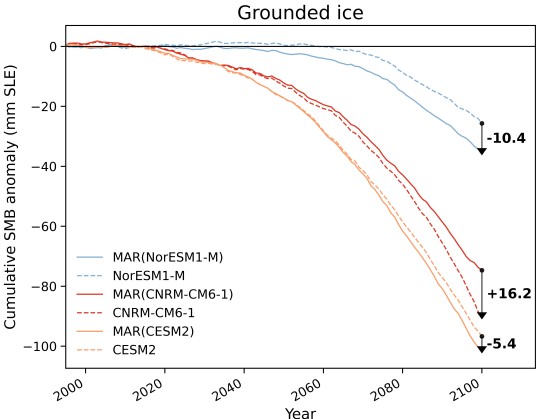

**Figure 11.** Cumulative contribution of the grounded Antarctic SMB (mmSLE) of MAR forced by NorESM1-M, CNRM-CM6-1, and CESM2 (solid ligne), and the ISMIP6-SMB directly computed from NorESM1-M, CNRM-CM6-1, and CESM2 (dashed lines) over 1995–2100. The differences between the cumulative contributions of the MAR experiments and their forcing ESM is also indicated in the figure.

Figure 10 compares future SMB changes (2081–2100 versus 1995–2014 i.e., the ISMIP6 reference period) projected by
MAR and the respective forcing ESMs. While the MAR projections are relatively insensitive to the forcing ESM for the same warming (see Sect. 3.2), the comparison between MAR and the forcing ESM reveals large differences, independently of the differences due to the higher resolution used in MAR that enables to distinguish high-elevation positives anomalies from low-elevation negative anomalies. For example, CNRM-CM6-1 projects a strong near-surface Antarctic warming (Fig. 1 and Nowicki et al. (2020), Fig. 1a) but the related runoff increase is particularly weak (Fig. S13), leading to only very slightly
negative anomalies, in contrast to MAR(CNRM-CM6-1), that simulates widespread negatives anomalies around nearly all the peripheral ice shelves consistent with a stronger warming 6. As highlighted by Fettweis et al. (2020), this suggests that the physics of the models and/or the biases over the current climate (in particular for the melt) could strongly influence the projected near-surface changes for identical changes in the free atmosphere. These MAR and ESM differences also highlight the importance of correctly representing the current climate and the need of additional projections relying on more models,
including both RCMs and ESMs. As the integrated differences cumulated over 1995–2100 can be larger or as large as the differences between CMIP5-RCP8.5 and CMIP6-ssp585, or between CMIP6-ssp126 and CMIP6-ssp245 (Fig. 11), this also raises the question of the sensitivity to the forcing of ISMIP6 projections, where the SMB is used as an input for performing projections of the total AIS mass balance (Seroussi et al., 2020).

## 4.3   Limitations

Our projections suggest a significant ablation by runoff as the firn would not absorb all the additional liquid water, whereas almost all surface meltwater refreezes in the snowpack. MAR does not include a liquid-water routing scheme that could either create liquid-water flowing over the ice surface, or accumulate melted water into surface or sub-surface lakes, further away



than the place of its production. The current view suggests that enhanced melt will be stored in crevasses or ponds that weaken ice shelves, potentially leading to their collapses by hydrofracturing (Scambos et al., 2000; Vieli et al., 2007; Pattyn et al., 2018). However, in some conditions, streams and rivers can transfer surface meltwater laterally and export it into the ocean (Kingslake et al., 2017; Pattyn et al., 2018; Dell et al., 2020; Arthur et al., 2020), which might eventually reduce the risk of hydrofracturing (Bell et al., 2017). Lake formation and meltwater runoff therefore represent a large uncertainty about the future of the ice shelves and the contribution of the AIS to sea level. These processes have yet to be implemented in the snowpack module of MAR.

MAR is not coupled to an ice-sheet model in these simulations and therefore has a static ice-sheet geometry (ie., fixed surface elevation and ice/ocean mask) that could lead to biases in the simulated SMB. For instance, melt-elevation feedback due to the lowering of the surface elevation by atmospheric and basal melt are not taken into account with a fixed geometry. Hence, we probably underestimate surface melt rates by overestimating the future surface elevation (Ritz et al., 2015). Since the ice/ocean mask is fixed over the whole simulation period (1975–2100), integrated anomalies could also be biased. The ice-shelf area and associated negative SMB value are potentially overestimated due to the absence of collapse processes. In the same way, the extent of grounded ice is reduced as grounding lines retreat, which should induce a negative contribution of the surface to sea-level rise. These two implicit consequences of using fixed ice mask and elevation could partly compensate each other. The elevation feedback has been shown to matter for 21st century Greenland projections (Le clec'h et al., 2019), but its importance for the AIS remains an open question.

Our simulations also do not include drifting snow, which can be active up to 81% of the time in some locations (Amory, 2020). Drifting snow has been simulated as the main present ablation component of the Antarctic ice sheet (Lenaerts and Van den Broeke, 2012; Van Wessem et al., 2018) and can lead to exposure of low-albedo and blue-ice area (Lenaerts et al., 2017a). The sublimation of eroded particles also cools the atmosphere (Le Toumelin et al., 2020(in preperation)) and has a significant influence on the humidity budget of the near-surface atmosphere (Amory and Kittel, 2019). The drifting-snow scheme of MAR had not yet been evaluated at the scale of the ice sheet when we performed our simulations and was therefore deactivated. Projected runoff ablation is much higher than present wind-driven ablation suggesting that drifting snow would not remain the main ablation process by 2100. This highlights the importance of assessing the future Antarctic drifting-snow climate in the global warming context.

## 5 Conclusions

In this study, we use the regional atmospheric model MAR, which includes determinant polar surface physics, forced by four carrefuly-selected ESMs (ACCESS1.3, NorESM1-M, CNRM-CM6-1, CESM2) to study the future evolution of the Antarctic SMB. These CMIP5 and CMIP6 models projects a wide range of Antarctic near-surface warming (+3.2°C to +8.5°C) and enable us to investigate the AIS sensitivity to different warmer climates in 2100.

Our results reveal an increase in grounded SMB (+390 $\mathrm{Gt\,yr^{-1}}$ to +745 $\mathrm{Gt\,yr^{-1}}$) between 1981–2010 and 2071–2100 due to an increase in snowfall amounts, despite higher runoff values partly offset this increase (up to 34%). Higher surface meltwater




production over the ice shelves at the end of the 21st century prevents a total absorption of additional liquid water by the snowpack, leading to high runoff values and mostly negative SMB anomalies. The spread over the ice shelves is however large, since our simulations project relatively stable SMB anomalies ( +28 $\mathrm{Gt\,yr^{-1}}$) to strong negative anomalies (-335 $\mathrm{Gt\,yr^{-1}}$). Our results suggest significant differences at the end of the century at the scale of the entire ice sheet, whether we consider the

grounded ice or ice shelves. However, future spatial and integrated changes for a same warming are similar, suggesting than uncertainties are mainly due to the sensitivity of ESMs to anthropogenic forcing and the timing of the projected warming.

Future changes modelled by MAR are strongly correlated with the near-surface warming of the forcing ESMs around the AIS. Using a statistical regression, we reconstruct integrated SMB anomalies over the grounded ice sheet as well as over the ice shelves for the whole CMIP5 (RCP8.5) and CMIP6 (ssp126, ssp245, ssp585) database. Over 2071–2100 compared to

the present, this reconstructed grounded SMB suggests a higher increase for CMIP6-ssp585 (+447 $\pm$ 134 $\mathrm{Gt\,yr^{-1}}$) than for CMIP5-RCP8.5 (+ 353 $\pm$ 114 $\mathrm{Gt\,yr^{-1}}$) that respectively corresponds to a 2000-2100 cumulated sea-level contribution of -6.3 $\pm$2.0 cm SLE and -5.1 $\pm$1.9 cm SLE. Low (ssp126) and intermediate (ssp245) CMIP6 emission scenarios project a lower negative contribution to sea-level rise than ssp585 (-3.0 $\pm$1.4 cm SLE using ssp126 and -4.2 $\pm$1.6 cm SLE using ssp245. Conversely, CMIP6-ssp585 yield a stronger SMB decrease over the ice shelves (-119 $\pm$ 100 $\mathrm{Gt\,yr^{-1}}$) than CMIP5-RCP8.5

(-54 $\pm$ 55 $\mathrm{Gt\,yr^{-1}}$).

Future SMB estimates are also used as forcing for ice-sheet models, notably in the ISMIP6 project, where SMB estimates are directly derived from ESMs. Despite several improvements in the latest generation of CMIP6 ESMs, using these models to study the evolution of the SMB involves several compromises that could lead to large uncertainties in the future SMB. We therefore compare the MAR projected SMB to the ISMIP6-derived SMB, revealing large local and integrated differences between

MAR and the respective forcing ESM. These MAR and ESM differences highlight the importance of correctly representing the current climate and the need of additional projections relying on more models including both RCMs and ESMs.

Under the Paris Agreement (limiting global warming to +1.5°C compared to pre-industrial temperature, which is a colder target than the projected mean CMIP6-ssp126 warming), increased surface melt over the ice shelves should remain weak, limiting potential ice-shelf collapses due to hydrofracturing. This weak increase in melt amounts should also limit surface

thinning and then positive feedbacks between surface damages and ice-shelf instability. However, large uncertainties remain in the influence of surface melt on the ice-shelf stability. Furthermore, our results highlight a warming threshold (+2.5°C) where the ice-shelf SMB could decrease, suggesting a low range of warming before potential irreversible damages on the ice-shelves. Finally, our simulations also suggest a stabilisation or even a decrease in grounded SMB with a +7.5°C near-surface warming, which would lead to a decrease in the sea-level mitigation capacity of the grounded AIS surface. This warming is however

reached before 2100 by only one model in the highest-emission scenario suggesting that more work is needed to asses the confidence of this threshold, the response of the AIS to strong warming after 2100 and AIS contribution to global sea-level rise.



*Code and data availability.* The MAR code used in this study is tagged as v3.11.1 on https://gitlab.com/Mar-Group/MARv3.7. See http://www.mar.cnrs.fr for more information about downloading MAR. The MAR SMB forced by ERA5 is available on the /ftp/climato/ckittel/MARv3.11/MAR-
ERA5. All the yearly files presented in this study - MAR simulations forced by reanalyses and ESMs - will be made available on https://zenodo.org after acceptance. Other higher-frequency results are also available upon request by email (ckittel@uliege.be).

*Author contributions.* CK, CAm, and XF designed the study. CK ran the simulations, made the plots (benefiting from some CAg and SH's scripts), performed the analysis and wrote the manuscript. CAm, CAg, NJ and XF provided importance guidance while all the authors discussed and revised the manuscript.

*Competing interests.* The authors declare that they have no conflict of interest.

*Acknowledgements.* We acknowledge the World Climate Research Programme's Working Group on Coupled Modelling, which is responsible for CMIP, and we thank the climate modelling groups for producing their model output and making it available. We also acknowledge ISMIP6 for making its datasets available. We appreciate the support of the University of Wisconsin-Madison Automatic Weather Station Program for the data set, data display, and information, NSF grant number 1924730, as well as all the institutes that collected data over the
ice sheet.

Computational resources have been provided by the Consortium des Équipements de Calcul Intensif (CÉCI), funded by the Fonds de la Recherche Scientifique de Belgique (F.R.S. – FNRS) under grant no. 2.5020.11 and the Tier-1 supercomputer (Zenobe) of the Fédération Wallonie Bruxelles infrastructure funded by the Walloon Region under grant agreement no. 1117545. This work was supported by the Fonds de la Recherche Scientifique – FNRS under grant no. T.0002.16.



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
