# Peer review of "Diverging future surface mass balance between the Antarctic ice shelves and grounded ice sheet"

_The Cryosphere, 2020_

## Referee Comment (RC1) · Anonymous Referee #1 · 11 Jan 2021

This paper fills an important gap left in the current research literature on future changes to the Antarctic Ice Sheet. My understanding of the current state of affairs is as follows. At present, centurial-scale predictions of future Antarctic ice volume use the outputs of global-scale earth system models to provide fields such as surface mass balance and runoff to ice flow models. These ESMs are necessarily run at a spatial resolution too coarse to resolve features like the Transantarctic Mountains. These topographic features would affect the local mass balance through feedbacks between elevation and precipitation and thus may be important for the purposes of ice flow modeling. Downscaling using a regional climate model such as MAR can provide the necessary resolution. Nowicki et al. 2020, due to time constraints and the computational cost of running RCMs, had little choice but to use ESM output directly to force ice sheet

models in their work. The present paper fills the gap by downscaling the outputs from several ESMs using MAR for Antarctica over this forcing period.

I'm an ice flow modeler, so I found the methodology convincing but would not be in a great position to critique it in the first place. The big takeaway that I got from this paper was the necessity of doing the right thing and using downscaled RCM output to force future ice flow projections rather than the less expensive approach of using ESM output directly. Figure 10 was particularly striking in illustrating the difference, especially for CNRM-CM6-1 in the vicinity of the Amundsen Sea Embayment and over the Siple Coast Ice Streams.

A few numbers are stated without additional context that might be helpful. For example, the authors state that the amount of precipitation falling as rain over the ice shelves will increase, but give only anomalies. Here it might be nice to say roughly what the total amount of rainfall is so that readers can get a feel for what the relative change is, or state that quantity directly.

Although it isn't strictly necessary, it would help to say something about what the oceans will do. It would be enough to add a single sentence stating that, while higher atmospheric temperatures and thus SMB over Antarctica may offset some sea level rise, increases in ocean heat content delivered to the ice shelves are likely to be a strong influence as well. You could cite Holland et al. 2019, West Antarctic ice loss influenced by internal climate variability and anthropogenic forcing. The authors do mention the possibility of other internal feedback mechanisms leading to ice shelf retreat, but I found the omission of any mention of ocean forcing to be quite glaring.

Overall I recommend for publication with a few minor corrections.

Technical corrections:

14: predict -> predicted

22: resultant -> result, or maybe sum

TCD
28: retaining -> restraining

59: the abbreviation ESM is used before the term "Earth System Model"

79: abundantly -> frequently or predominantly

114: Why not REMA or BedMachine? Was this for consistency with work from before those products were available?

195: contrasted -> contrasting

332: cumulated -> summed or aggregated

343: "Although RCMs have been believed..." -> Some studies have argued that RCMs..."

355: "that simulates..." -> "which simulates"

360: cumulated -> aggregated

396: carrefuly -> carefully

TCD

---

## Referee Comment (RC2) · Anonymous Referee #2 · 19 Jan 2021

**Summary**

In this paper, Kittel and co-authors present a series of experiments in which they use the regional climate model MAR to simulate climate over Antarctica over the coming century with boundary conditions provided by four different earth system models (drawn from CMIP5 and CMIP6). They find a substantial difference in the surface mass balance of grounded versus floating ice in all cases, with the former acting as a net sink for sea level, and the latter acting as either a contributor or neutral, depending on the forcing. The authors also find that the integrated differences between MAR response to different ESMs is largely explained by differences in the timing and intensity of projected global warming. This allows the authors to develop simple polynomial

functions relating near-surface temperature anomaly in the Antarctic region to SMB, snowfall, rainfall, and runoff anomalies. They then apply these relationships to the remainder of the CMIP5 and CMIP6 ensemble members to produce an approximation of model uncertainty in SMB anomaly for each CMIP scenario.

I find this paper to be a well-written and useful contribution to the community's understanding of variability in climate model predictions. It does a good job of laying out critical assumptions and also is careful to couch their results as model predictions (rather than a factual future). Besides a few requests for changes to the structure of the paper, and a few technical corrections, I suggest that the paper be published with little further review.

**Major Points**

I believe that Supplement S2 should be included in the main text, more or less in its entirety. The results section's primary points are devoted to summarizing its content and referencing its figures, so why not just include it in the manuscript?

I also believe that it would be appropriate to cross-validate the quadratic fits with an independent dataset. For example, if one fits this polynomial to 3 of the 4 experiments, how well is the fourth predicted? Showing that it does a good job would go a long way towards ensuring that this surrogate model (which is what it is) is likely to be skillful at predicting the anomalies for other models.

**Minor Points**

**L13** Specify what a 'lower surface mass balance' means. More negative?

**Fig. 1** Would it be possible to include observed temperature anomaly in some form? maybe ERA5?

**L128** It's okay to leave the details to the references, but it would be helpful to at least qualitatively describe the methodology for comparing ESMs to ERA5 that are used here.

**L170** SIC: 'Suplement'

**L227** SIC: 'questionning', SIC: 'in pace'

**Sec 4.1** When reporting the bounds on SLE in this section, is the error in the surrogate model accounted for? The fit to the simulations isn't perfect, so there should be some extra variance injected to account for potential mismatch between Eq. 2/3 and the true model predictions, rather than just the spread in the predictions themselves.

---

## Author Comment (AC1) · 25 Jan 2021

We first would first like to thank reviewer #1 for making constructive comments which will help to improve our manuscript. Please find our answers to your comments in blue in the text.

This paper fills an important gap left in the current research literature on future changes to the Antarctic Ice Sheet. My understanding of the current state of affairs is as follows. At present, centurial-scale predictions of future Antarctic ice volume use the outputs of global-scale earth system models to provide fields such as surface mass balance and runoff to ice flow models. These ESMs are necessarily run at a spatial resolution too coarse to resolve features like the Transantarctic Mountains. These topographic features would affect the local mass balance through feedbacks between elevation and precipitation and thus may be important for the purposes of ice flow modeling. Downscaling using a regional climate model such as MAR can provide the necessary resolution. Nowicki et al. 2020, due to time constraints and the computational cost of running RCMs, had little choice but to use ESM output directly to force ice sheet models in their work. The present paper fills the gap by downscaling the outputs from several ESMs using MAR for Antarctica over this forcing period.

I'm an ice flow modeler, so I found the methodology convincing but would not be in a great position to critique it in the first place. The big takeaway that I got from this paper was the necessity of doing the right thing and using downscaled RCM output to force future ice flow projections rather than the less expensive approach of using ESM output directly. Figure 10 was particularly striking in illustrating the difference, especially for CNRM-CM6-1 in the vicinity of the Amundsen Sea Embayment and over the Siple Coast Ice Streams.

Thank you for these remarks and for pointing out the importance of choosing the right method.

A few numbers are stated without additional context that might be helpful. For example, the authors state that the amount of precipitation falling as rain over the ice shelves will increase, but give only anomalies. Here it might be nice to say roughly what the total amount of rainfall is so that readers can get a feel for what the relative change is, or state that quantity directly.

We have preferred to give values relative to present rather than raw values for two main reasons:
-although our simulation has been  successfully evaluated with respect to the available observations, the integrated values of our reference simulation may not be correct
-even if our downscalings of each ESM are very close to our reference simulation (see table S2), they remain not unbiased.

 We therefore wanted to put more emphasis on the changes rather than on the raw values, by not highlighting the potential biases over present climate (and probably stationary in future following Krinner et al. (2019)).

However, we understand the reviewer's remark and propose to add the following tables with the raw values over grounded and ice-shelf SMB and components over the period 2071-2100 in additional material:

| Mean (Gt yr$^{-1}$) over 2071–2100 | SMB | Snowfall | Rainfall | Net Sublimation | Runoff | Melt |
|---|---|---|---|---|---|---|
| Grounded ice (11.94 10$^6$ km$^2$) | | | | | | |
| MAR(ACCESS1.3) | 2596 | 2829 | 50 | 117 | 166 | 330 |
| MAR(NorESM1-M) | 2573 | 2688 | 27 | 103 | 39 | 108 |
| MAR(CNRM-CM6-1) | 2829 | 3084 | 98 | 87 | 266 | 527 |
| MAR(CESM2) | 2909 | 3148 | 83 | 97 | 225 | 419 |
| Ice shelves (1.77 10$^6$ km$^2$) | | | | | | |
| MAR(ACCESS1.3) | 372 | 637 | 57 | 55 | 267 | 536 |
| MAR(NorESM1-M) | 584 | 694 | 29 | 55 | 84 | 253 |
| MAR(CNRM-CM6-1) | 180 | 679 | 124 | 49 | 574 | 869 |
| MAR(CESM2) | 263 | 706 | 97 | 48 | 492 | 781 |

We also propose to add  (L176):

*Projected SMB and components values are given compared to their respective mean values over current climate to remove the dependence of the potential linear biases over current climate but raw values over the grounded ice sheet and ice shelves are available in Supplement (Tab.~S3).*

Although it isn't strictly necessary, it would help to say something about what the oceans will do. It would be enough to add a single sentence stating that, while higher atmospheric temperatures and thus SMB over Antarctica may offset some sea level rise, increases in ocean heat content delivered to the ice shelves are likely to be a strong influence as well. You could cite Holland et al. 2019, West Antarctic ice loss influenced by internal climate variability and anthropogenic forcing. The authors do mention the possibility of other internal feedback mechanisms leading to ice shelf retreat, but I found the omission of any mention of ocean forcing to be quite glaring

We thank the reviewer for this suggestion which will certainly help to put our results into a wider context.

We will modify L31-33 :
*Since the 2000s, the Antarctic ice sheet has been losing mass at an accelerating rate mainly due to an increased ice discharge in the West AIS (Shepherd et al., 2018), itself caused by the acceleration of outlet glaciers in response to basal (ocean) melt thinning the ice shelves and reducing their buttressing effect (Paolo et al., 2015; Gardner et al., 2018; Rignot et al., 2019).*

by

*Since the 2000s, the Antarctic ice sheet has been losing mass at an accelerating rate mainly due to an increased ice discharge in the West AIS (Shepherd et al., 2018), itself caused by*

*the acceleration of outlet glaciers in response to basal (ocean) melt thinning the ice shelves and reducing their buttressing effect (Paolo et al., 2015; Gardner et al., 2018; Rignot et al., 2019). Stronger basal melting of ice shelves is further projected to drive future Antarctic mass loss (Holland et al., 2019; Serroussi et al., 2020).*

We will also change the conclusion L422-424:

*Under the Paris Agreement (limiting global warming to +1.5°C compared to pre-industrial temperature, which is a colder target than the projected mean CMIP6-ssp126 warming), increased surface melt over the ice shelves should remain weak, limiting potential ice-shelf collapses due to hydrofracturing.*

by

*Although other processes such as basal melting of ice shelves could lead to their disappearance (Holland et al., 2019; Seroussi et al., 2020), increasing surface melt should remain weak, limiting potential ice-shelf collapses due to hydrofracturing under the Paris Agreement (limiting global warming to +1.5°C compared to pre-industrial temperature, which is a colder target than the projected mean CMIP6-ssp126 warming).*

28: retaining -> restraining 59: the abbreviation ESM is used before the term "Earth System Model" 79: abundantly -> frequently or predominantly 114: Why not REMA or BedMachine? Was this for consistency with work from before those products were available? 195: contrasted -> contrasting 332: cumulated -> summed or aggregated 343: "Although RCMs have been believed..." -> Some studies have argued that RCMs..." 355: "that simulates..." -> "which simulates" 360: cumulated -> aggregated 396: carrefuly -> carefully

Thanks for all the technical corrections that we included in our manuscript.

Bedmap was used instead of a more recent topography dataset (REMA or BedMachine) to enable comparison of the results from MARv3.11 (this study) to MARv3.6 Agosta et al (2019) in the aim of facilitating the model development. At the time of Agosta et al (2019), these datasets were not yet available. However, updating the topography dataset in MAR is one of the next planned developments.

References in this answer:

Agosta, C., Amory, C., Kittel, C., Orsi, A., Favier, V., Gallée, H., van den Broeke, M. R., Lenaerts, J., van Wessem, J. M., van de Berg, W. J., et al.: Estimation of the Antarctic surface mass balance using the regional climate model MAR (1979-2015) and identification of dominant processes, Cryosphere, 13, 281–296, 2019.

Krinner, G. and Flanner, M. G.: Striking stationarity of large-scale climate model bias patterns under strong climate change, Proceedings of the National Academy of Sciences, 115, 9462–9466, 2018.

---

## Author Comment (AC2) · 25 Jan 2021

We first would like first to thank reviewer #2 for making constructive comments which will help to improve our manuscript. Please find our answer in blue in the text.

In this paper, Kittel and co-authors present a series of experiments in which they use the regional climate model MAR to simulate climate over Antarctica over the coming century with boundary conditions provided by four different earth system models (drawn from CMIP5 and CMIP6). They find a substantial difference in the surface mass balance of grounded versus floating ice in all cases, with the former acting as a net sink for sea level, and the latter acting as either a contributor or neutral, depending on the forcing. The authors also find that the integrated differences between MAR response to different ESMs is largely explained by differences in the timing and intensity of projected global warming. This allows the authors to develop simple polynomial functions relating near-surface temperature anomaly in the Antarctic region to SMB, snowfall, rainfall, and runoff anomalies. They then apply these relationships to the remainder of the CMIP5 and CMIP6 ensemble members to produce an approximation of model uncertainty in SMB anomaly for each CMIP scenario.

I find this paper to be a well-written and useful contribution to the community's understanding of variability in climate model predictions. It does a good job of laying out critical assumptions and also is careful to couch their results as model predictions (rather than a factual future). Besides a few requests for changes to the structure of the paper, and a few technical corrections, I suggest that the paper be published with little further review.

**Major Points**

I believe that Supplement S2 should be included in the main text, more or less in its entirety. The results section's primary points are devoted to summarizing its content and referencing its figures, so why not just include it in the manuscript?

As section S2 is only an evaluation of the MAR results on the present, we preferred to put it as supplementary material in order not to lengthen the manuscript and lose the main thread of the story. Nevertheless, following the reviewer's remark, we propose to move Fig. S3 and Tab. S2 into the main manuscript. We will also create a section before Section results based on previous Section S2 and that assesses the SMB downscalings over the present period (while SMB-component and near-surface-climate evaluation figures would remain in the supplement):

[revised manuscript text omitted]

I also believe that it would be appropriate to cross-validate the quadratic fits with an independent dataset. For example, if one fits this polynomial to 3 of the 4 experiments, how well is the fourth predicted? Showing that it does a good job would go a long way towards ensuring that this surrogate model (which is what it is) is likely to be skillful at predicting the anomalies for other models.

We cross-validated our "regression model" fitted on 3 experiments compared to the last remaining one (See Fig. R1 to R4).  Although the results remain good, the remaining error reflects the inherent error of the regression linked to the simplification, but also the inter-model variability for a same temperature threshold which cannot be captured by the other 3 models. For example, Figure R3B shows that the regression based on ACCESS1.3, NorESM1-M and CESM2 underestimates the negative anomaly of SMB on ice shelves. This is expected because the regression does not take into account the very negative anomalies that only MAR(CNRM-CM6-1) does. This also highlights the importance of inter-model variability for the projections and thus for maximising the projected warming.

The RMSE of each cross-validation can be compared to the RMSE of the original "validation". The cross-reconstruction is better when compared to ACCESS1.3 (NorESM1-M) over the grounded ice sheet (over the ice shelves) and of the same order over the ice

shelves (the grounded ice sheet). When compared to CNRM-CM6-1 or CESM2, the RMSE is larger. As these experiments simulate larger anomalies, the same relative error leads to a larger absolute error. Furthermore, as explained before, these "cross-regression" do not take into account all the inter-model (or inter-downscaling) variability inducing a bias. This is especially true for the strongest warmings, which are reached by only two models (CNRM-CM6-1 and CESM2). Removing one of these two during the crossfit therefore results in a large increase in uncertainty (and error) for the strongest anomalies caused by the strongest warming as it precludes the representation of model inter-variability. This again highlights the importance of using as many as ESM candidates as possible and that more downscalings are needed to reduce projected uncertainties.

We think that this cross-validation gives similar conclusions to the evaluation of our regression already present in additional material so we suggest to not add it.

[Figure]

Fig R1: Evaluation of the MAR(ACCESS1.3) reconstructions based with regressions derived from the three other model anomalies (f(x) = -3.2 TAS² + 130.5 TAS -17.3 and f(x) = -13.4 TAS² + 37.4 TAS - 3.8) compared to the original MAR SMB anomalies over the grounded ice (A) and the ice shelves (B).

[Figure]

Fig R2: Evaluation of the MAR(NorESM1-M) reconstructions based with regressions only derived from the three other model anomalies (f(x) = -0.7 TAS² + 109.6 TAS - 3.3 and f(x) = -12.2 TAS² + 29.1 TAS  - 3.8) compared to the original MAR SMB anomalies over the grounded ice (A) and the ice shelves (B).

[Figure]

Fig R3: Evaluation of the MAR(CNRM-CM6-1) reconstructions based with regressions only derived  from the three other model anomalies (f(x) = -0.1 TAS² + 115.2 TAS  -12.6 and f(x) = -10.3 TAS² + 27.1 TAS - 1.4) compared to the original MAR SMB anomalies over the grounded ice (A) and the ice shelves (B).

[Figure]

Fig R4: Evaluation of the MAR(CESM2) reconstructions based with regressions only derived from from the three other model anomalies (f(x) = -2.5 TAS² + 111.8 TAS -12.1 and f(x) = -15.9 TAS² + 40.1 TAS - 5.7) compared to the original MAR SMB anomalies over the grounded ice (A) and the ice shelves (B).

**Minor Points**

L13 Specify what a 'lower surface mass balance' means. More negative?

We suggest to change L13:
*Over the ice shelves, the strong runoff increase associated with higher temperature is projected to lower the SMB with a stronger decrease in CMIP6-ssp585 compared to CMIP5-RCP8.5.*

by:

*Over the ice shelves, the strong runoff increase associated with higher temperature is projected to decrease the SMB (more strongly in CMIP6-ssp585 compared to CMIP5-RCP8.5).*

Fig. 1 Would it be possible to include observed temperature anomaly in some form? maybe ERA5?

We added ERA5 over 1960 -- 2020 using the latest backward release of ERA5 prior to 1979. However, initial results suggest that the reanalysis is not as reliable as before 1979 than after (Hersbach et al., 2020). This is likely a combination of climatic isolation of Antarctica and poor observation coverage. The new Fig.1 now also illustrates that ESMs correctly reproduce the mean warming since 1960.

[Figure]

*Fig1.: Time series of the 90°S–60°S annual near-surface temperature anomaly (°C) between compared to the reference period (1981–2010) from the ERA5 reanalysis and ESMs using the extreme high-emission scenarios RCP8.5 and ssp585 after their historical period (2004 for CMIP5 and 2014 for CMIP6). The thick blue and red lines represent the mean annual warming from 28 CMIP5 and 34 CMIP6 ESMs. Thinner orange and blue lines are for ESMs selected as boundary conditions for our regional climate model MAR: CNRM-CM6-1 and CESM2 (CMIP6, ssp585) and NorESM1-M and ACCESS1-3 (CMIP5, RCP85). The dashed black line is the ERA5 reanalysis (1960--2020).*

L128 It's okay to leave the details to the references, but it would be helpful to at least qualitatively describe the methodology for comparing ESMs to ERA5 that are used here.

This aspect of the paper had been less developed as it was based entirely on the method defined in Agosta et al, 2015 and Barthel et al, 2020. In addition, the extended discussion including CMIP6 of this selection will be the subject of a paper (Agosta et al., in preparation) with in part the same authors of our manuscript. It will include the rankings of each mode including new CMIP6 models (more models are now available than when we had to make our selection) and a discussion on the influence of the choice of metrics and other parameters not included in the score (resolution too low, potential importance of some metrics in relation to others) and on the strategies to be followed when selecting models (i.e. take only the best on the present climate, diversify warming or exclude models with too large a ECS even if this means excluding some good models) for both the Greenland and Antarctic ice sheets. We felt that this whole subject deserved to be discussed separately as it could have an important influence on the projections that will be made and the possible impacts of these results on policy makers.

However, we have extended the section presenting the selection procedure (L121-128) following the reviewer's remarks. Furthermore we have inverted the two first paragraphs of the selection section .

We changed:

*The selection of ESMs that were dynamically downscaled by MAR was based on their ability to 1) represent the current climate (air temperature and humidity, sea surface conditions,*

*and large-scale circulation) around the AIS and 2) diversify the projected changes during the 21st century. These criteria ensure on one hand, that the ESM biases will not have a prejudicial effect on the projections since the present state determines future biases (Agosta et 2015, Krinner and Flanner 2018) and on the other hand that we assess the AIS response to a wide range of projected temperature increases for a better quantification of the future uncertainties. We therefore selected ESMs by comparing them to the ECMWF reanalysis ERA5 (Hersbach et al., 2020) over the recent "historical" period (1980--2004) following the method defined in Agosta et al., 2015 and Barthel et al. (2020) for CMIP5, extended here to CMIP6 and applied only to the Antarctic atmosphere.*

*Large-scale forcing models were chosen among the CMIP5 and CMIP6 ESMs. CMIP6 models rely on an improved and more sophisticated representation of the global climate system than CMIP5. They incorporate better coupling between the different components of the Earth system, improved present- and better-constrained future concentrations scenarios of long-lived greenhouse gases and aerosols (Eyring et al., 2016,O'Neill et al., 2016). Additionally, most CMIP6 ESMs are also run on a higher spatial resolution. First analyses of the CMIP6 results revealed higher equilibrium climate sensitivity in this new-generation models (Mauritsen et al., 2019, Voldoire et al., 2019, Zelinka et al., 2020, Meehl et al., 2020, Wyser et al., 2020), suggesting warmer future climates, while based on similar future scenarios in terms of global radiative forcing. However, this higher climate sensitivity is potentially not supported by paleo-climate records (Zhu et al., 2020). We therefore also included models from the CMIP5 dataset, some of which show a good agreement with reanalyses over the current Antarctic climate (Agosta et al., 2015, Palerme et al., 2017). We only chose the scenarios of large greenhouse gas emissions from CMIP5 (RCP8.5) and its updated version in CMIP6 (ssp585) in order to obtain  stronger warming signals. These two scenarios have an equivalent global radiative forcing of +8.5 W m-2 by 2100, but differ in how the anthropogenic forcing is split between individual drivers of global warming (O'Neill et al., 2016).*

by
*Large-scale forcing models were chosen among the CMIP5 and CMIP6 ESMs. CMIP6 models rely on an improved and more sophisticated representation of the global climate system than CMIP5. They incorporate better coupling between the different components of the Earth system, improved present- and better-constrained future concentrations scenarios of long-lived greenhouse gases and aerosols (Eyring et al., 2016,O'Neill et al., 2016). Additionally, most CMIP6 ESMs are also run at a higher spatial resolution. First analyses of the CMIP6 results revealed higher equilibrium climate sensitivity in this new-generation models (Mauritsen et al., 2019, Voldoire et al., 2019, Zelinka et al., 2020, Meehl et al., 2020, Wyser et al., 2020), suggesting warmer future climates, while based on similar future scenarios in terms of global radiative forcing. However, this higher climate sensitivity is potentially not supported by paleo-climate records (Zhu et al., 2020). We therefore also included models from the CMIP5 dataset, some of which show a good comparison with reanalyses over the current Antarctic climate (Agosta et al., 2015, Palerme et al., 2017). We only chose the scenarios of the largest greenhouse gas emissions from CMIP5 (RCP8.5) and its updated version in CMIP6 (ssp585) in order to obtain stronger warming signals and then SMB sensitivities. These two scenarios have an equivalent global radiative forcing of +8.5 W m-2 by 2100, but differ in how the anthropogenic forcing is split between individual drivers of global warming (O'Neill et al., 2016).*

*The selection of ESMs that were dynamically downscaled by MAR was based on their ability to i) represent the current climate (air temperature and humidity, sea surface conditions, and large-scale circulation) around the AIS and ii) diversify the projected changes during the 21st century. These criteria ensure on one hand, that the ESM biases will not have a prejudicial effect on the projections since the present state determines future biases (Agosta et al., 2015, Krinner and Flanner 2018) and on the other hand that we assess the AIS response to a wide range of projected temperature increases for a better quantification of the future uncertainties for a same scenario. We therefore firstly ranked ESMs by comparing them to the ECMWF reanalysis ERA5 (Hersbach et al., 2020) over the recent "historical" period (1980--2004) following the method defined in Agosta et al. (2015) and Barthel et al. (2020) for CMIP5, extended here to CMIP6 and applied only to the Antarctic atmosphere. The method firstly computes the root mean square error (RMSE) compared to ERA5 for several climate variables (mean air temperature at 850 hPa, annual precipitable water, annual sea level pressure, summer sea surface temperature and winter sea ice extent over 1980--2004) that are supposed to determine the SMB (Agosta et al. , 2015). The score of each ESM is then obtained by averaging its RMSEs that were previously normalized with regards to the multi-model median and interquartile range. This enables the combination of several metrics using the same weight for each of the metrics. Once the models were ranked on the basis of their score against ERA5, the final selection was made to diversify the changes expected at the end of the century and the availability of 6-hourly outputs in the CMIP5/CMIP6 database at the end of 2019 when we started our experiments.*

L170 SIC: 'Suplement'
L227 SIC: 'questionning',
SIC: 'in pace'
Thanks for highlighting these mistakes! They were corrected in the revised version of the manuscript.

Sec 4.1 When reporting the bounds on SLE in this section, is the error in the surrogate model accounted for? The fit to the simulations isn't perfect, so there should be some extra variance injected to account for potential mismatch between Eq. 2/3 and the true model predictions, rather than just the spread in the predictions themselves

The regression error has not been included in the shading of Figures 8 and 9 or the bounds since we only aimed to illustrate the uncertainty caused by inter-model variability. We recognize that the total future uncertainty should be greater if we add to the inter-model uncertainty the uncertainty due to the intrinsic errors of the regression, in a way that is difficult to quantify.

Nevertheless, the RMSEs between the original and reconstructed anomalies (ie, 68 kg/m²yr and 38 kg/m²yr ) are 1) lower (a little more larger) for the grounded (ice shelves) than the present SMB variability 2) lower than the future projected changes in our simulations or in the CMIP reconstructions 3) lower than the interannual variability in MAR projected anomalies 4) lower than the inter-model variability within a same scenario.

The regression error thus appears to be of lower importance than present and future interannual variability, projected changes and inter-model variability, suggesting that it introduces at least second-order uncertainty with respect to all these indicators.

We will add after L338:
*Note that the uncertainties associated to mean reconstituted anomalies are only based on the intermodel variability over both the grounded ice sheet and the ice shelves but the uncertainties would have been larger if the biases of MAR (over current climate) and Eq.~2 and Eq. ~3 had been taken into account.*